# A Cascaded Adaptive Network-Based Fuzzy Inference System for Hydropower Forecasting

**DOI:** 10.3390/s22082905

**Published:** 2022-04-10

**Authors:** Namal Rathnayake, Upaka Rathnayake, Tuan Linh Dang, Yukinobu Hoshino

**Affiliations:** 1School of Systems Engineering, Kochi University of Technology, 185 Miyanokuchi, Tosayamada, Kami 782-8502, Kochi, Japan; hoshino.yukinobu@kochi-tech.ac.jp; 2Department of Civil Engineering, Faculty of Engineering, Sri Lanka Institute of Information Technolog, Malabe 10115, Sri Lanka; upaka.r@sliit.lk; 3School of Information and Communications Technology, Hanoi University of Science and Technology, No. 1, Dai Co Viet Road, Hanoi 100000, Vietnam; linh.dangtuan@hust.edu.vn

**Keywords:** Cascaded-ANFIS, GRU, regression, LSTM, RNN, Sri Lanka, hydropower, forecasting

## Abstract

Hydropower stands as a crucial source of power in the current world, and there is a vast range of benefits of forecasting power generation for the future. This paper focuses on the significance of climate change on the future representation of the Samanalawewa Reservoir Hydropower Project using an architecture of the Cascaded ANFIS algorithm. Moreover, we assess the capacity of the novel Cascaded ANFIS algorithm for handling regression problems and compare the results with the state-of-art regression models. The inputs to this system were the rainfall data of selected weather stations inside the catchment. The future rainfalls were generated using Global Climate Models at RCP4.5 and RCP8.5 and corrected for their biases. The Cascaded ANFIS algorithm was selected to handle this regression problem by comparing the best algorithm among the state-of-the-art regression models, such as RNN, LSTM, and GRU. The Cascaded ANFIS could forecast the power generation with a minimum error of 1.01, whereas the second-best algorithm, GRU, scored a 6.5 error rate. The predictions were carried out for the near-future and mid-future and compared against the previous work. The results clearly show the algorithm can predict power generation’s variation with rainfall with a slight error rate. This research can be utilized in numerous areas for hydropower development.

## 1. Introduction

The Sustainable Development Goals (SDGs) were announced in 2012. Seventeen goals were recommended to be completed by 2030. One of the essential aims at the list is to achieve clean energy generation [1]. Global hydropower output peaked in 2020 with 38.2 exajoules, up from 37.7 exajoules the previous year, and climbed by 11.6 exajoules in the two decades from 2000 to 2020 [2]. Thus, hydropower contributes more than 16% of total energy generation [3]. Many South Asian nations, including Sri Lanka, fulfill a considerable portion of their electrical demand through hydropower facilities (approximately 40% of total energy in Sri Lanka) [4]. Renewables are still regarded as being one of the most environmentally friendly power producing systems in the world. As a result, a 75–100% increase in production capacity is projected in the coming years [3]. In comparison to wealthy countries, which have utilized 70% of their total capacity, emerging nations have only built 23% of financially feasible hydropower plants [5]. As a result, many developing nations are rapidly spending considerable resources in developing hydropower facilities, since they are seen as safe and cost-effective sources of renewable energy that minimize carbon emissions [6].

Hydropower is one of the cleanest forms of energy sources; however, the inflow to dam reservoirs significantly impacts the pace of hydropower output. Therefore, hydropower generation is very unpredictable due to its dependency on meteorological conditions and weather conditions. Furthermore, climate change is likely to disrupt hydropower plant operations by unbalancing the water cycle, increasing the frequency of rainfall events, and rising atmospheric temperatures. It is evident that the evaporation and other water cycle components are affected by the predicted temperature change of 0.0164 °C annually [7]. Rainfall, on the other hand, is projected to increase in some countries while decreasing in other countries, thereby impacting hydropower producing capacity [8].

If electricity output is dramatically curtailed due to the negative consequences of climate change, the hydropower sector might become one of the most vulnerable businesses. In addition, water scarcity in the catchments and reduced hydropower generation inputs due to landslides or soil erosion might exacerbate the problem. On the other hand, construction of hydroelectric infrastructure is prohibitively expensive, presents substantial dangers to the aquatic ecology, and produces socioeconomic concerns [9].

As a result, forecasting hydropower output is critical for maximizing renewable energy consumption to meet growing the demand and control hydroelectric power management. This would help to achieve environmental sustainability. Despite this, estimating future hydropower output is challenging due to the nonlinearities of the input functions and regional and temporal fluctuations in meteorological data, including temperature and rainfall. As a result, the prediction output of a good model might provide a substantial financial benefit by regulating renewable energy infrastructure development, such as hydroelectric infrastructure [10].

## 2. Related Works

Several researchers have studied the impacts of climatic fluctuations on hydroelectric output, primarily utilizing Global/Regional Climate Models (GCMs/RCMs), predictive modeling, and conventional statistical methodologies (e.g., [11,12,13]).

Several methods to predict the future of hydropower plants using machine learning techniques can be found in the literature, and ANN is one of the main algorithms that can be used to carry out this task. A case study was carried out in Nigeria, Jebba, and Kainji, employing ANN impartial input data [14]. In Uzlu et al. [15], the artificial bee colony method was used to forecast future hydropower output throughout Turkey utilizing input factors such as generation capacity, energy consumption, population, and temperatures. According to the report, the power output of Turkey is not in accordance with the country’s objective of producing 30% of its electricity through renewable means in 2023. Furthermore, Patil [16] estimated future streamflow for the Ranganadi River, which is located in India, up to 2040, to forecast hydropower output using three GCM models and ANN. When using feed-forward back-propagation algorithms of ANN architecture, input parameter characteristics substantially influence forecasting future power generation [17].

Furthermore, while projecting electricity output from various energy resources in the United States, Khodaverdi [18] proposed an ANN–ARIMA hybrid model rather than ANN to predict future renewable energy generation (e.g., hydroelectricity, solar, and wind). After examining 66 studies that used ANN to improve reservoir operations, the study by Ajala et al. [19] further reinforced the idea of combining ANN with supervised or unsupervised learning algorithms to improve reservoir outflow prediction. Furthermore, the study by Shaktawat and Vadhera [5] advised performing further research on risk management in hydropower utilizing a fuzzy model mixed with ANN and genetic algorithm.

Some scientists insist that ANNs are important in hydropower predictions. Anuar et al. showcased that the hidden layer neurons had a more significant impact on the results of the ANN structure when forecasting streamflow at The Malaysian hydroelectric dam [20]. Furthermore, Sessa et al. [21] discovered that ANN models are the most accurate at predicting short-term and long-term hydropower generation after having conducted research studies on run-of-the-river (ROR) hydroelectricity in France, Portugal, and Spain using chronological weather information such as rainfall, snow, and temperature.

However, the related research in the context of Sri Lanka is minimal. In fact, per the authors’ knowledge, only one such study was performed in Sri Lanka that used ANN to anticipate electricity output. Furthermore, the research by Karunathilake and Nagaha [22] estimated daily electricity consumption but did not forecast power generation.

Although numerous ANN-based machine learning algorithms have been found in the literature for hydropower prediction, machine learning techniques that use fuzzy logic to predict hydropower generation are handful. Some of the literature on fuzzy logic-based predictions are listed in the following paragraphs.

The Grey wolf approach was combined with an adaptive neuro-fuzzy inference system (ANFIS) to anticipate hydroelectricity generation in Dehghani et al. [23]. In addition, hydropower output of Albania was analyzed by Konica and Staka [24] to establish the best forecasting model for assessing hydro energy production for the years 2007–2016. They have used the fuzzy time series approach to forecast Albania’s hydropower generation.

Moreover, some studies have been conducted to forecast the rainfall using fuzzy logic-based algorithms. Rainfall forecasting was performed in a study by Suprapty et al. [25] of the East Kalimantan area, which has 13 watersheds with the potential for a micro-hydropower plant. To simulate rainfall time series data, the auto-regressive (AR) model based on a fuzzy inference system (FIS) was utilized. The research work done by Rahman et al. [26] showcased an improvement to forecast rainfall using a fuzzy rule-based approach. Eight distinct equations were created using temperature, wind velocity, and precipitation. The minimum content of the induction component in temperature and wind velocity fuzzifications was investigated, as were fuzzy levels and membership functions.

Mostly, time-series predictions are purely non-linear, and fuzzy logic is the best of artificial intelligence for tackling non-linear problems [27].

The majority of the earlier works share the following flaws.

Generally, artificial neural network-based algorithms are bulky in the complexity of the calculations.The methods are to use when the predictions depend on the uncertainty factors and non-linear inputs.The methods are not likely to generate the best possible predictions because the input factors vary depending on the different environments.The methods are require enormous amounts of computing power.

Therefore, while addressing the above-mentioned overall flaws, this study tries presents a new algorithm called cascaded adaptive neuro-fuzzy inference system (Cascaded ANFIS) to predict the hydropower generation [28]. The impacts of this research can be pointed out as follows.

This system uses fuzzy logic approach along with a neural network to address the uncertainty and the non-linearity of the inputs.The base algorithm of this system is two-input one-output ANFIS, and the computational power reduces dramatically.It is possible to generate a near-zero error in the prediction by increasing the number of levels in the Cascaded ANFIS algorithm.This study presents future power generation up to the year 2099 using two different climate models.The comparative study presented in this work provides a solid understanding of the potential regarding the Cascaded ANFIS algorithm compared to that of the cutting-edge time series prediction algorithms.

### Hydropower in Sri Lanka

Sri Lanka has a hydroelectric power potential of 1719 megawatts (MW), and existing hydropower growth pledges could contribute around 247 MW to the power grid in the coming decades [4]. According to Gunasekara [29], the bulk of Sri Lanka’s hydroelectric plants are more than 25 years old. Although hydropower plants have lifespans of about 50 years, if any of the older hydroelectric dams fail to operate, whether due to climate or mechanical faults, Sri Lanka will have energy shortages, because it would be challenging to replace defective hydroelectric dams in a brief period [30].

As a result, in the Sri Lankan context, analyzing the power generating capabilities of its hydroelectric projects is crucial. In doing so, one must consider this developing country’s economic electrical demands, and the management of water supply infrastructure development amid climatic factors. Several analyses in Sri Lanka, however, have looked at potential energy production from current or planned hydroelectric dams. The study in Udayakumara et al. [31] looked at ways to increase power output in hydroelectric dams by preventing land degradation and reservoir floods in the Uma Oya valley, one of Sri Lanka’s most crucial significant catchment areas.

The study in Chandrasekara et al. [2] studied inflows in the Kotmale reservoir until 2005 from 1960 using the El Nino Southern Oscillation (ENSO) phase indicator and discovered that flow to the basin had decreased, impacting hydropower output and agricultural plans. According to the research in Imbulana et al. [32], a rise in continuous rainfall events, a decrease in continuous dry weather, and a gain in yearly rainfall series will improve the future production capacity of the Mahaweli watershed’s hydropower plants.

Khaniya et al. [12] used a multiyear rainfall trend study to demonstrate that changes in climate will have no effect on Denawaka Ganga mini-hydropower, as in the Rathnapura area. The study released in Perera and Rathnayake [33] additionally sought to analyze the effect of climate change on the Erathna mini-hydropower station in the Rathnapura area. They concluded that electricity generation will decline in the following years.

The study in Khaniya et al. [34] undertook a similar evaluation on the recently operational Uma Oya watershed, and the researchers found that there will be no substantial challenges to hydroelectric generation or the groundwater limits in the years ahead in the watershed region.

As stated in the introduction, there seems to have been no comprehensive study on hydroelectric forecasts in Sri Lanka for the coming decades. Consequently, this study has a better possibility of attracting the attention of the Sri Lankan authorities than most, to enhance the management and forecasting procedures in hydroelectric plants.

## 3. Study Area

The Samanalawewa Hydropower Project is located in the central portion of Sri Lanka, in the Belihul Oya region of Rathnapura division, Sabaragamuwa province. The project was completed in 1992, just downstream of the confluence of Belihul Oya to Walawe River. The watershed region (359 km^2^) is midland, made of marble and quartz, and has an average altitude of around 530 m [30]. The region is located inside the rainy region of the country (wet zone), having a mean annual precipitation of around 2500 mm [35]. The southwest monsoon provides the majority of the rainfall for the catchment, though there are minor contributions from the northeast monsoon and inter-monsoon storms. The Samanalawewa Hydroelectric Project includes a U-shaped rockfill dam which is around 110 m high from its foundation. The power station is capable of producing 124 MW as per the design guidelines. Figure 1 illustrates a detailed catchment map.

Samanalawewa hydroelectric is among Sri Lanka’s oldest and one of the largest reservoir-type power stations, and has long played an essential part in maintaining power distribution stability during peak times. It accounts for 8.69% of the total power generated by the larger hydroelectric plants. Since its start, this project has aroused significant attention owing to the leakage problem discovered on the lake’s right bank due to poor geological characteristics [36]. Moreover, several environmental difficulties were noted during the design stage; however, few precautions were taken because no stringent environmental restrictions necessitated substantial development efforts [37].

Although the Environmental Impact Assessment (EIA) framework was established in Sri Lanka in 1988, EIA during the building of Samanalawewa was primarily centered on vegetation revascularization and habitat conservation.

Due to the apparent leak, phase 2 of the hydropower plant construction (120 MW capacity) was suspended; therefore, a mini-hydropower facility was constructed that utilities the leaking water. Despite the Ceylon Electricity Board’s (CEB’s) valiant efforts to halt the leak, stored water continues to flow at a pace of 2.1–2.8 m^3^/s [38].

Irrigated water from the dam is vital for agricultural usage in downstream settlements, such as Kaltota, Madabadda, Welipotayaya, and Koongahamankada. Paddy yields downstream of the study area have been reduced by 11.5 percent due to a lack of water in the reservoir [39]. Therefore, water management is highly important.

As a portion of the confiscated water is immediately delivered for irrigation without going through the power station, analyzing the prospective availability of water in the Samanalawewa dam for energy production is crucial. Another fraction (the leaking component) is supplied by mini-hydropower plants that produce far less energy. Furthermore, with the rising availability of water from downstream agricultural districts, water management at the Samanalawewa reservoir must be more carefully managed. Furthermore, climate variability may have an influence on CEB’s watershed management goals at the Samanalawewa hydroelectric station, either positively or negatively. As a result, the following study will be of interest to the many stakeholders of the Samanalawewa Hydropower Project.

To assess that, the monthly rainfall data were purchased from the Department of Meteorology, Sri Lanka for the rainfall stations showcased in Figure 1. The data were collected from 1992 to 2018 as per the availability. There are some missing data due to various reasons, including instrumentation errors. Therefore, the data were screened carefully before they were used. Balangoda, Alupola, Detanagalla, Belihuloya, Nonpareil (Belehuloya), and Nagrak Estate are the six stations which were used in this study. The descriptive statistics of the dataset are shown in Table 1.

## 4. Methodology

The overall explanation of the method used in this study is presented in this section. The development process is several steps. Initially, futuristic climate data were extracted and corrected their biases using the linear bias correction technique. Then, the Cascaded ANFIS algorithm was used to generate the outputs for each pair of inputs. This process is explained in the algorithm usage subsection.

Furthermore, three state-of-the-art algorithms, namely, GRU, RNN, and LSTM, were used to distinguish the efficiency of the algorithms.

### 4.1. Climate Data Extraction for Future

Global Climatic Models (GCMs) accommodate climatic data at vast ranges across immensely different landscapes. In contrast, Regional Climatic Models (RCMs) are employed at more inadequate orders and can accommodate more specific data for adaptation evaluation, and preparation [40]. As projecting instruments, GCMs forecast the climate variance of the Earth in the future. They should, however, be investigated on a local or even global scale to identify efficient correspondence procedures.

Future climatic data for various situations can be retrieved. Such scenarios are known as Representative Concentration Pathways (RCP), in which weather data can be obtained. RCPs can be expressed as trajectories on the Intergovernmental Panel on Climate Change’s [41] greenhouse gas concentrations. RCP 2.6, 4.5, 6.0, and 8.5 are the four most generally applied RCPs in the literature [41]. RCP4.5 is the intermediate emission scenario, in which emissions begin to decline around 2045. RCP8.5 is the leading emission situation, in which discharges proceed to rise during the 21st century.

It is generally known that RCMs have variable degrees of methodical bias [42,43]. The causes of such preferences could be due to methodical model mistakes produced through poor conceptualization, spatial averaging, and discretizations in grid cells. Some prejudice improvement strategies have been employed in the literature to address these biases [44]. Linear scaling, local intensity scaling, power transformation, variance scaling, distribution transfer, and delta change approach are some widely used techniques for removing biases in climatic data.

The linear scaling (LS) approach [45] is employed extensively in various investigations due to its simplicity and speed of application. LS can adjust all-climate elements to an appropriate level; however, few examples of precipitation corrections can be found—see Gimire et al., Lafon et al., Luo et al., and Mahmood et al. [46,47,48,49]. The bias correction method for linear scaling can be implemented employing the two equations provided here (Equations (Equation 1) and (Equation 2)), where his, cor, sim, obs, *d*, and *P* stand for raw RCM data, bias-corrected data, raw RCM corrected data, observed data, daily, and precipitation, respectively, and *m* is the long-term cyclical average of rainfall data: (1)$$P_{his,d}^{cor}=P_{his,d}*\frac{\mu_{m}\left(P_{obs,d}\right)}{\mu_{m}\left(P_{his,d}\right)}$$
(2)$$P_{sim,d}^{cor}=P_{sim,d}*\frac{\mu_{m}\left(P_{obs,d}\right)}{\mu_{m}\left(P_{sim,d}\right)}$$

LS technique was used to remove the biases in the RCP precipitation products, as shown in the Equations (Equation 1) and (Equation 2). The ground measured monthly rainfalls were used to remove these biases.

#### 4.1.1. Implementation of the Cascaded ANFIS Algorithm

ANFIS is a hybrid algorithm that incorporates two different methods, a neural network (NN) and fuzzy logic (FL). As a result, in ML, ANFIS has both the benefits of NN and FL [28]. ANFIS is a six-layer structure, the first layer being the input and the final layer being the output. The membership functions are constructed in the second layer using FL. The third layer generates the cumulative product of the previously generated membership function. The following layer defuzzifies the outputs from the third and fourth levels before feeding them to the final layer, which generates the output.

ANFIS, on the other hand, takes absolute values as inputs and transforms them into fuzzy values. The fuzzy reasoning is then generated based on the membership functions and rules. After that, the fuzzy values are transformed to crisp values [50]. The Cascaded ANFIS algorithm is a repeatable ANFIS implementation with two primary inputs and one output. Figure 2 depicts the creation of this algorithm. This approach can be used in conjunction with ANFIS because iterations can route the answer to be more accurate than the ANFIS algorithm with five layers.

The critical difference between the Cascaded ANFIS algorithm and the conventional ANFIS algorithm is that the product of the standard ANFIS algorithm fits the input of the conventional ANFIS method’s subsequent usage. However, fuzzy is applied as the fuzzification process within the ANFIS model’s internal layers, just as in the traditional ANFIS technique. The usage of membership functions, which change numerical values into fuzzy members, is used to achieve fuzzification. The pair selection technique and the training method are the two main components of the Cascaded ANFIS algorithm.

The pair selection module tackles the first significant issue with ANFIS. The usual method is to decrease the input dimensions before applying an algorithm. On the other hand, the unique approach applies every feature to construct a sturdy model, which may be helpful for noisy datasets. The revolutionary Cascaded ANFIS algorithm’s training module deals with computational complexity. The combination selection method employs sequential feature selection (SFS). This approach is unusual because it identifies the most suitable match for individual input variables using a 2-input, 1-output ANFIS structure.

In the training method, the 2-input ANFIS structure is again employed. As the input variables are linked to the most suitable match from the former method, they can be immediately fed into the ANFIS module, which will generate current outputs and RMSE for specific data combinations. There is also a pre-determined goal error at this time, and the RMSE is then compared to the anticipated error as a result. The procedure can be terminated if the target error is fulfilled. If not, the algorithm moves on to the next iteration. This document for implementation [28] has a detailed description of the Cascaded ANFIS algorithm, including pseudo-code.

As mentioned in the above sections on dataset generation for future rainfall, six data points were generated for every month from the year 2021 to the year 2099 using RCP 4.5 and RCP 8.5 climate models. Accordingly, these four data points were used as the inputs to the Cascaded ANFIS algorithm. As shown in Figure 3, Balangoda, Alupola, Detanagalla, Belihuloya, Nonpareil, and Nagrak Estate were the inputs to the first level of the Cascaded ANFIS algorithm. Each input was coupled with the best pair because the ANFIS structure is a two-input one-output configuration. The process of the paring of each input is discussed in detail in the pair selection section of this paper [28]. $ANFIS_{2}^{1}$ is a two-input one-output ANFIS module. As shown in the figure, at each iteration level, individual six $ANFIS_{2}^{1}$ modules are used to generate separate outputs from the pairs.

As pointed out in the figure, there are six outputs from level one. Then, the second level will initialize by applying those outputs as inputs to the second level. Again, the pair selection process is performed to select the best pair for each ANFIS in the second level. This process continues until the pre-defined maximum level is reached. In the end, the outputs are averaged to find the final value *f* (Equation (Equation 3)). Here, $O_{n,j}$ is the output of the *j*th ANFIS module at level *n*.
(3)$$f=\frac{\sum_{j=1}^{6}O_{n,j}}{6}$$

Furthermore, the dataset was divided in to training and testing as 70% and 30% in this study, and we used the same data piles for all the algorithms.

#### 4.1.2. Parameter Settings for Each Algorithm

This study was conducted to investigate the best prediction algorithm among ours and the state-of-the-art algorithms in hydropower forecasting. Hence, we used several algorithms, and each algorithm was created with the optimum parameters. The following is the complete list of algorithms used in this study.

Multilayer Perception (MLP)K-Nearest Neighbors (KNN)Adaptive Network-based Fuzzy Inference System (ANFIS)Particle Swarm Optimization with ANFIS (ANFIS-PSO (Hybrid))Genetic algorithms with ANFIS (ANFIS-GA (Hybrid))Linear regressionLasso regressionRidge regressionRecurrent neural network (RNN)Long short-term memory (LSTM)Gated recurrent unit (GRU)Cascaded ANFIS

Here, two types of algorithms were used: general machine learning algorithms and regression machine learning algorithms. MLP, KNN, and ANFIS methods can be considered as the general machine learning algorithms; and linear, lasso, ridge, LSTM, GRU, and RNN can be introduced as regression models.

Each algorithm was separately coded and run during the study to generate the outputs. Most of the algorithms’ parameters were manually adjusted, and some of the algorithms were adjusted under the considerations of other literature. Each parameter for each algorithm is shown in Table 2.

The experiment was carried out for the hydropower generation dataset. Nine different algorithms were tested, and the best algorithm was chosen based upon the Root Mean Square Error (RMSE) and the Coefficient of Determination ($R^{2}$) of each algorithm. The *RMSE* and $R^{2}$ can be calculated as shown in Equations (Equation 4) and (Equation 5).
(4)$$RMSE=\sqrt{\frac{1}{q}\sum_{t=1}^{q}{\left(\bar{u}\left(t\right)-\hat{u}\left(t\right)\right)}^{2}}$$
(5)$$R^{2}=1-\frac{RSS}{TSS}$$
where, in Equation (Equation 4), $\bar{u}\left(t\right)$ is the prediction and $\hat{u}\left(t\right)$ is the real output. *q* is the size of the population. In Equation (Equation 5), the sum of the squares of the prediction is RSS, and the sum of squares of real values is TSS.

## 5. Results and Discussion

This section includes two main subsections. First, the algorithm comparison is introduced, since selecting the best algorithm was one of the main objectives of this study. Second, the future power generation is explained alongside the results.

### 5.1. Comparison of the Algorithms

Table 3 presents the RMSE for each algorithm at the training and the testing phases. The smallest errors of 1.01 in the training and 1.80 in the testing were obtained by Cascaded ANFIS. As mentioned in the introduction of Cascaded ANFIS, the error reduces while propagating through levels. Hence, a higher level of structure generates more accurate results at the cost of computation. The results shown here are for Cascaded ANFIS at level 20.

Moreover, the second, third, and fourth best accuracies were achieved by LSTM, GRU, and RNN. They obtained 6.03, 6.50, and 7.85 errors in the training, sequentially. It is also worth remarking that the other ANFIS algorithms, such as ANFIS, ANFIS-PSO, and ANFIS-GA, presented higher error rates when compared with the other algorithms.

Furthermore, the Coefficient of Determination ($R^{2}$) was calculated for each algorithm, as shown in Figure 4 and Figure 5. Figure 4 shows the performances of general machine learning algorithms, and Figure 5 shows regression machine learning algorithms’ performances. $R^{2}$ is used to examine how variations in one variable may be explained by changes in another.

$R^{2}$ shows the percentage variance in y explained by x-variables. The measure runs from 0 to 1 (the x-variables can explain 0% to 100% of the variation in y).

The best $R^{2}$ was given by Cascaded ANFIS, 0.929. GRU, LSTM, and RNN had $R^{2}$ of 0.711, 0.701, and 0.634, respectively.

However, LR and Lasso Regression show a similar *R*^2^ which is 0.061 as in Figure 5. Here, the training and the testing of these algorithms were conducted using the same
experimental conditions. The calculation of the difference between the real value and the
prediction was conducted up to eight decimals. LR and Lasso Regression calculation results
were almost the same except for the last few decimals. When presenting the results in this
paper, the accuracies were rounded up to two decimals points, and it caused the plots of
LR and Lasso Regression to be the same in this analysis.

The increase in $R^{2}$ of the Cascaded ANFIS by level can be seen in Figure 6. For level 1, $R^{2}$ is 0.422 because only two variables are considered the input to ANFIS modules at the first level. Then, at level 10, the $R^{2}$ value increases by almost 50%. Finally, at level 20, the value reaches almost 1 (0.929). Therefore, this result explains that Cascaded ANFIS outperforms all other algorithms used here, including regression models. Hence, Cascaded ANFIS was used to forecast hydropower generation up to the year 2099.

### 5.2. Forecasting of Hydropower Generation in the Future

Figure 7 showcases the projected power generation for the near future under the RCP4.5 and RCP8.5 climate scenarios. It can be seen herein that both climate scenarios have projected significant declination of power generation by Samanalawewa Hydropower Plant. The declination is monotonic except for a couple of years’ slight inclinations. However, interestingly, the power generation in RCP4.5 is lower than that of RCP8.5. Many development projects are expected in Sri Lanka, and they will require significant amounts of power. Around a 1000 MW power demand is projected for Sri Lanka in the future. In addition, Sri Lanka has proposed to generate more than 70% of its power demand using renewable resources by the 2030s. However, the Samanalawewa power plant’s results for the near future do not support the requirements in the near future. This is critical, as this power plant significantly contributes to Sri Lanka’s power demand via a renewable resource.

Figure 8 presents the projected power generation for mid-future years from both RCP scenarios. Unlike in the near future, the projected power generation patterns have zig-zag patterns for both climatic scenarios, but they still showcase overall declining trends. In addition, the significant differentiation in the projected power generation from RCP4.5 and RCP8.5 for the near future cannot be seen in the mid-future, and instead, an overlap of the climatic scenarios can be seen.

Nevertheless, the projected power generation under RCP4.5 and RCP8.5 climatic scenarios showcases the impact of climate change on the hydropower generation in a healthy hydropower plant in Sri Lanka. Even though Figure 7 and Figure 8 present the annual power generation, seasonal impacts can also be seen at higher resolution scales, such as monthly power generation. According to such results, climate change will adversely impact Samanalawewa Hydropower Plant in the near future and mid-future, even though Sri Lanka’s power demand is in escalating phase. Therefore, the findings of this research can be used for critical discussions by the stakeholders and then enhance the countermeasures.

Clear differences can be seen for the power generation predictions with the two different techniques (Figure 7 and Figure 9). Khaniya et al. (2020) [12] have used frequently used ML algorithms via ANNs. Significant reductions can be seen for RCP4.5 using the Cascaded ANFIS algorithm. Therefore, the results have to be carefully assessed with time. The analysis can be restructured in the short-term.

Figure 8 illustrates the projected power generation from 2041 to 2099. A similar illustration to mid-future (2041–2070) power generation can be seen for the far future (2071–2099) too. However, the projections overall do not showcase declining or inclining trends, even though they have peaks and troughs. Nevertheless, as per the authors’ understanding, it is too early to comment on power generation in the far future. RCP scenarios have projections for the far future; however, the high variability of climate and its relationship to greenhouse gas emissions might change the future patterns. In addition, the world’s green energy plans, such as electric vehicles, should positively impact the changing climate in the long run. Even though we have projected power generation for the far future, quick conclusions may not be feasible.

## 6. Conclusions

Hydropower generation for Samanalawewa Hydropower Plant was forecasted using a novel Cascaded ANFIS algorithm under RCP4.5 and RCP8.5 for future years. The accuracy of the newly utilized algorithms is higher compared to other frequently used algorithms. It has shown lower RMSEs and higher $R^{2}$. The authorities would be interested in the prediction model due to it’s robustness for the practical applications. However, the algorithm takes some significant time to train the forecasting model. The future projection is interesting. The projection was considered for the near future and mid future cases based on the design life of a hydropower station. Therefore, the suggestions for future forecasting should align with the design life of the hydropower plant. Replacement of various important instrumentation like turbines can significantly influence the efficiency of the power generation. Therefore, the results presented herein are based on the system which is currently available. Based on these, the model can successfully be utilized to forecast power generation for future years. Thus, the authorities and planners can learn the future generation and then to matches the required demand. In addition, the authorities can make decisions regarding replacements of various instrumentation to enhance the efficiency of the Samanalawewa hydropower station. Nevertheless, the results are somewhat contrasting to the results presented by Khaniya et al. (2020) [12]. Therefore, a detailed analysis should be carried out with time to state sound conclusions.

## Figures and Tables

**Figure 1 sensors-22-02905-f001:**
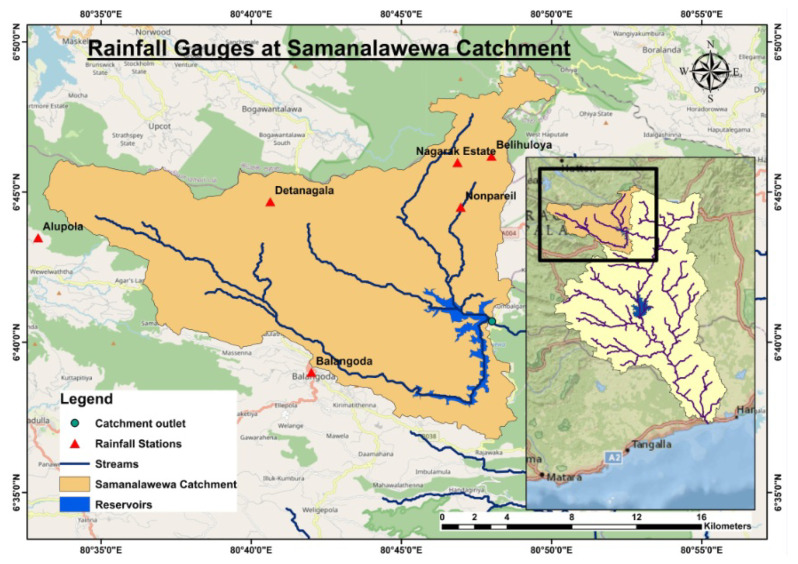
Rainfall gauges at Samanalawewa catchment.

**Figure 2 sensors-22-02905-f002:**
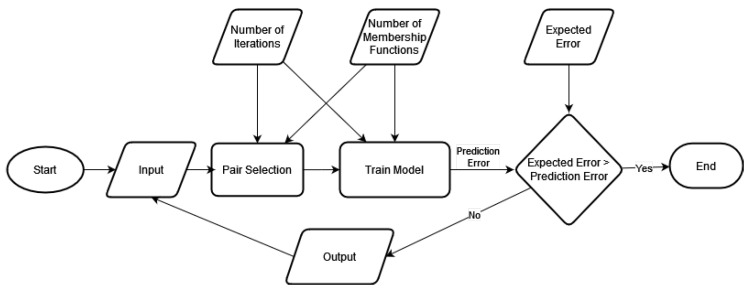
Flowchart of the Cascaded ANFIS.

**Figure 3 sensors-22-02905-f003:**
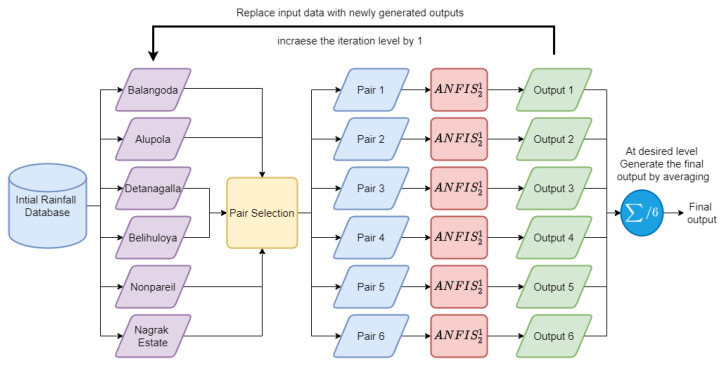
Hydropower prediction Cascaded ANFIS structure.

**Figure 4 sensors-22-02905-f004:**
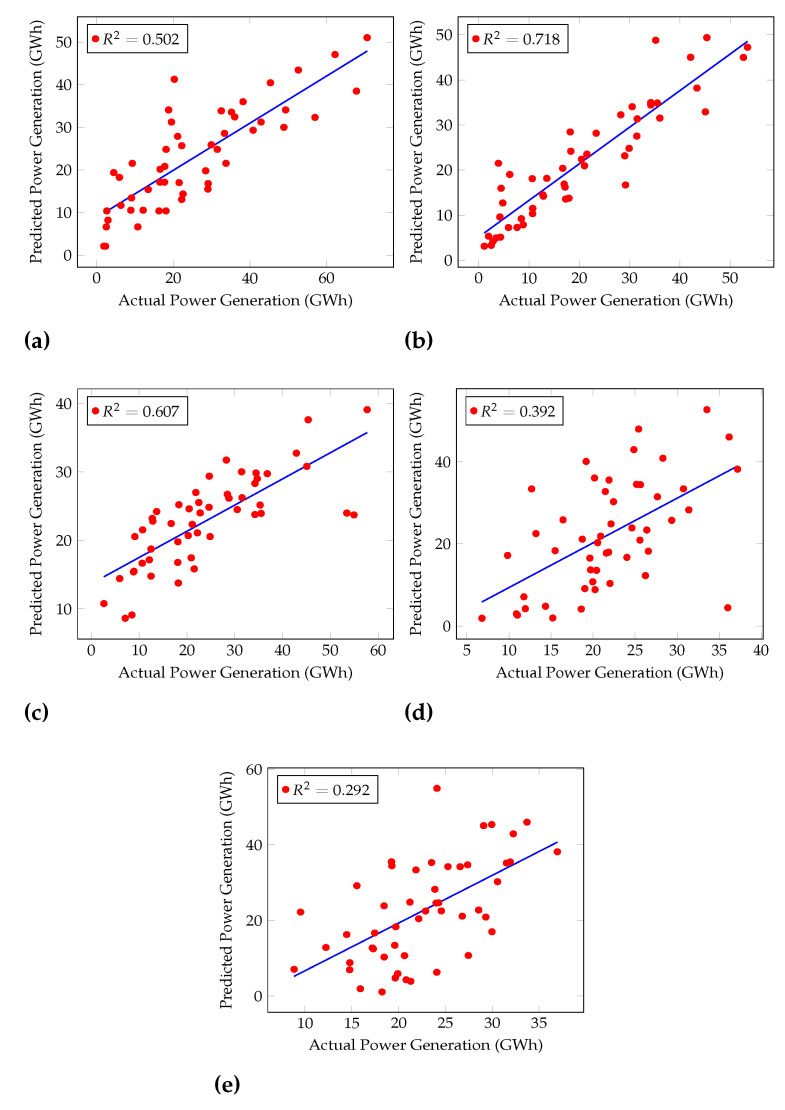
Coefficients of Determination $\left(R^{2}\right)$ of Rain Fall Test dataset for (**a**) KNN, (**b**) MLP, (**c**) ANFIS (**d**) PSO-ANFIS, and (**e**) GA-ANFIS.

**Figure 5 sensors-22-02905-f005:**
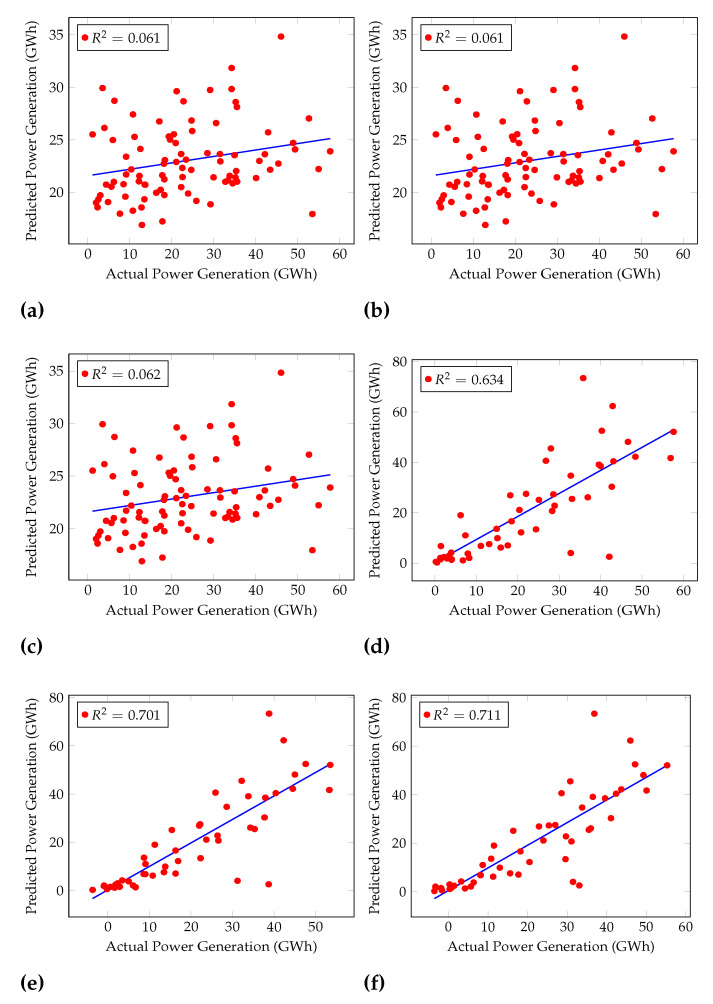
Coefficients of Determination $\left(R^{2}\right)$ of Rain Fall Test dataset for (**a**) linear regression, (**b**) lasso regression, (**c**) ridge regression (**d**) RNN, (**e**) LSTM, and (**f**) GRU.

**Figure 6 sensors-22-02905-f006:**
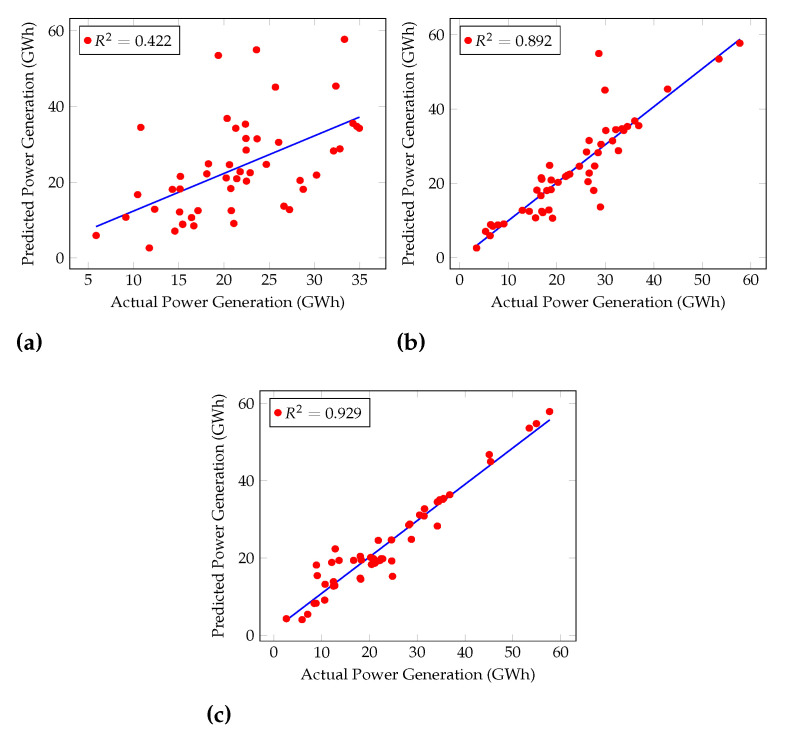
Cascaded ANFIS behavior for different levels. (**a**) Level 1, (**b**) level 10, (**c**) level 20.

**Figure 7 sensors-22-02905-f007:**
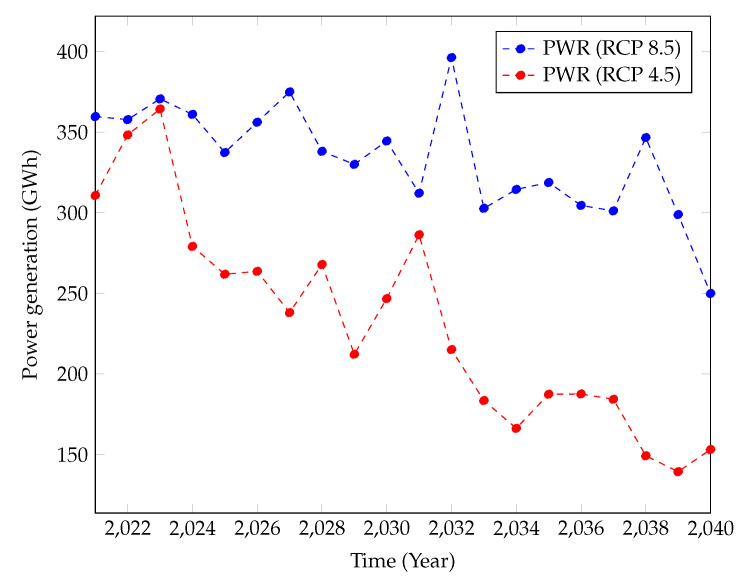
Power generation predictions from year 2021 to 2040.

**Figure 8 sensors-22-02905-f008:**
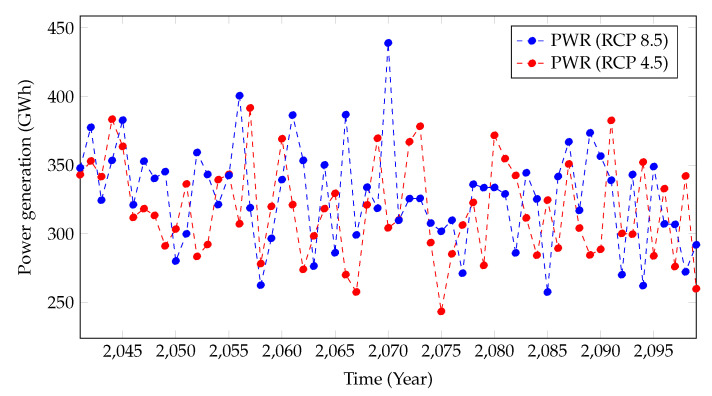
Power generation predictions from 2041 to 2099.

**Figure 9 sensors-22-02905-f009:**
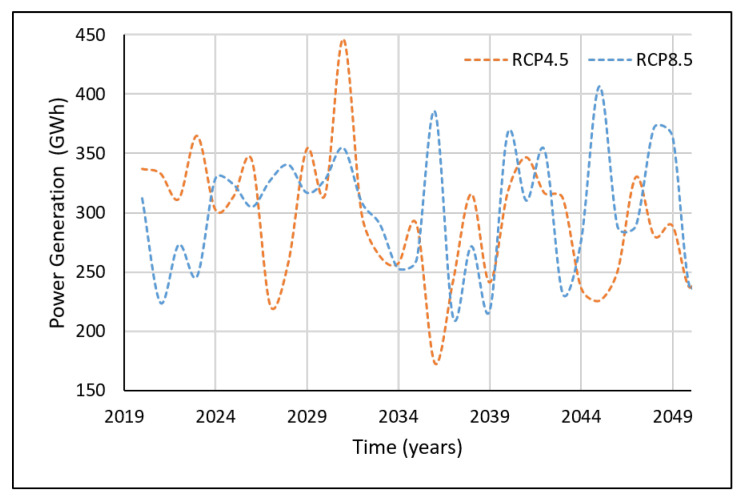
Hydropower predictions from Khaniya et al. (2020) [12].

**Table 1 sensors-22-02905-t001:** Descriptive statistics of the system.

	Balangoda	Alupola	Detanagalla	Belihuloya	Nonpareil	Nagrak Estate	Power
count	127.00	127.00	127.00	127.00	127.00	127.00	127.00
mean	377.88	190.57	221.81	240.77	183.42	187.65	22.86
std	224.50	161.46	215.17	218.55	156.57	183.21	14.69
min	27.40	7.50	0.00	2.70	0.00	0.67	1.10
25%	205.35	61.35	50.55	83.30	54.54	40.31	10.72
50%	348.10	136.60	144.50	160.20	132.20	124.95	21.04
75%	509.05	308.05	349.90	353.95	289.53	282.10	34.00
max	1159.90	734.70	926.10	1371.00	661.30	930.30	67.85

**Table 2 sensors-22-02905-t002:** Parameter settings for each algorithm.

Algorithm	Parameters	
MLP	Hidden layer size	50, 50, 50
	Activation	tanh
	Solver	adam
	alpha	0.05
	learning rate	constant
KNN	Weights	Uniform
	n_neighbors	1
ANFIS	Iteration	100
	Membership Functions	3
	Step Size	0.1
	Decrease rate	0.9
	Increase rate	1.1
ANFIS-PSO	Inertia Weight	1
	Inertia weight damping ratio	0.99
	Personal Learning Coefficient	1
	Global Learning Coefficient	2
ANFIS-GA	Crossover Percentage	0.7
	Mutation Percentage	0.5
	Mutation Rate	0.1
	Selection Pressure	8
	Gamma	0.2
RNN/LSTM/GRU	Optimizer	adam
	Learning rate	0.0001
	Activation	relu
	batch size	30
	epochs	100
Cascaded ANFIS	Iteration	100
	Membership Functions	3
	Step Size	0.1
	Decrease rate	0.9
	Increase rate	1.1

**Table 3 sensors-22-02905-t003:** RMSE for training and testing data.

Algorithm	*RMSE* (Train)	*RMSE* (Test)
MLP	7.52	25.26
KNN	9.73	19.33
ANFIS	10.47	18.06
ANFIS-PSO	10.99	16.61
ANFIS-GA	11.88	16.87
Linear Regression	13.74	14.85
Lasso Regression	13.72	14.82
Ridge Regression	13.70	14.88
RNN	7.85	11.62
GRU	6.50	8.33
LSTM	6.03	6.88
Cascaded ANFIS	1.01	1.80

## Data Availability

The data can be available only for acceptable research purposes from the authors.

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
