# Peer review of "A Cascaded Adaptive Network-Based Fuzzy Inference System for Hydropower Forecasting"

_sensors, 2022, doi:10.3390/s22082905_

Round 1
Reviewer 1 Report
This is an interesting paper, about the hydropower forecasting using fuzzy logic. The authors focus on cascaded ANFIS in order to model and forecast power generation from rainfall.
Positive remarks about the paper:
- It is well written.
- Previous related work is adequately referenced.
- It contains results, which might be really useful for researchers in the field, and especially those belonging to the natural resources community.
- The statistical assessment is solid.
- The proposed research is essential for the m environmental sciences.
- The performance of the proposed model is compared with a number of AI-ML algorithms
- The keywords accurately reflect the content.
Minor issues
- English language should be revised
- Fig 9 should be moved from conclusion
My overall impression is that the manuscript is suitable for publication after the submission of a slightly revised version of their manuscript, taking into consideration the above minor issues.
Reviewer 2 Report
This paper is well organised, and the studied idea is attractive. However, the following minor corrections need to be addressed:
1- There are a few grammar mistakes
2- Figures 9 and 10 must be presented before the conclusions
3- Some figures can be merged into one figure; for example, figures 8, 9 and 10 can be presented in one figure with a longer x-axis scale (this point is optional)
Reviewer 3 Report
In this article, using Cascaded Adaptive Network-Based Fuzzy Inference System and several other types of machine learning methods including MLP, KNN, etc. have been used to predict hydropower in Samanalawewa:
1. Line 218 indicates that data was collected over a period of 1992 to 2018. After modifying the data by the LS method, explain how the 150 properties (Figure 3) were reduced. Exactly how to reduce the feature ?. The results should be presented in two modes without reducing the feature and all of them.
2. The total number of data should be specified exactly and also how they are divided (training percentage and test percentage)?
3. Were the training and test data the same for all methods used? Explain?
4. Is KNN used for forecasting?
5. Why not use other methods such as RBF?
6. Although it is mentioned that the novel Cascaded ANFIS algorithm is not a new method in my opinion. Explain.
7. The article claims that the model was used to predict the years 2041 to 2070 and 2021 to 2040. The question that arises is how the features have been calculated for these years ???
8. How to really be confident for the year 2041. The results can be trusted for at least a few years after the last year.
9. State the innovation of your work clearly.
10. Describe your data in a table of descriptive statistics features.
